# Towards Stabilized Few-Shot Object Detection with Less Forgetting via Sample Normalization

**DOI:** 10.3390/s24113456

**Published:** 2024-05-27

**Authors:** Yang Ren, Menglong Yang, Yanqiao Han, Weizheng Li

**Affiliations:** School of Aeronautics and Astronautics, Sichuan University, Chengdu 610207, China; renyang@stu.scu.edu.cn (Y.R.); hanyanqiao@stu.scu.edu.cn (Y.H.); liweizheng@stu.scu.edu.cn (W.L.)

**Keywords:** few-shot learning, object detection, meta-learning

## Abstract

Few-shot object detection is a challenging task aimed at recognizing novel classes and localizing with limited labeled data. Although substantial achievements have been obtained, existing methods mostly struggle with forgetting and lack stability across various few-shot training samples. In this paper, we reveal two gaps affecting meta-knowledge transfer, leading to unstable performance and forgetting in meta-learning-based frameworks. To this end, we propose sample normalization, a simple yet effective method that enhances performance stability and decreases forgetting. Additionally, we apply Z-score normalization to mitigate the hubness problem in high-dimensional feature space. Experimental results on the PASCAL VOC data set demonstrate that our approach outperforms existing methods in both accuracy and stability, achieving up to +4.4 mAP@0.5 and +5.3 mAR in a single run, with +4.8 mAP@0.5 and +5.1 mAR over 10 random experiments on average. Furthermore, our method alleviates the drop in performance of base classes. The code will be released to facilitate future research.

## 1. Introduction

Object detection is a fundamental task in computer vision and has achieved particularly remarkable development with the help of large-scale annotated data sets [1,2,3] and well-designed powerful detectors [4,5,6,7,8,9,10]. However, in practical applications, the frequent occurrence of low-data scenarios limits the capability of detectors, making few-shot object detection (FSOD), which aims to solve such tasks with a small number of training samples, a critical approach in bringing object detection from theoretical research to real-world applications and deployments.

Many FSOD approaches have been developed to improve the generalization ability of neural networks, which can be mainly divided into transfer-learning-based [11,12,13,14,15,16,17] and meta-learning-based [18,19,20,21,22,23,24,25] methods. The former type aims to train suitable network parameters for invariant representation across domains and focuses on how to freeze fewer components of the detector without performance degradation [26,27]. Consistent with the general object detection framework, transfer-learning-based approaches present streamlined training pipelines without complex episodic procedures. However, such methods are limited by the few-shot setting to gain a map from the source domain to the target domain, which makes them lack generalization and robustness, facing serious overfitting and forgetting. In contrast, meta-learning usually consists of two branches, called support and query, and pays more attention to the aggregate information of these two branches to acquire the ability of ‘learning to learn’, which brings better robustness and generalization ability compared with transfer-learning-based methods.

However, we observe that meta-learning-based methods are generally unable to maintain consistent performance when trained on different data and may encounter the issue of forgetting. The detection results of meta methods vary widely, especially for lower-shot scenarios. As illustrated in Figure 1b, the classical meta-learning-based method Meta-DETR [18] exhibits instability when detecting a novel class, such as ‘cat’, failing to detect anything with the first two-shot training objects, and cannot distinguish between ‘dog’ and ‘cat’ with the third training samples. Most FSOD methods prioritize the performance for novel classes over a single run or average results over multiple experiments. Researchers of some methods [12,28] have noted the large sample variance due to few-shot samples and have focused on the worst performance under different samples, but they have not proposed an effective way to balance stability and accuracy. Additionally, meta-learning-based methods also suffer from the problem of forgetting, leading to poor performance in base classes especially under one-shot scenes, although several works [29] have attempted to address this problem with extremely complex architectures. We also note that for the object detection task, high-dimensional feature space makes the vectors suffer from the hubness problem [30,31], which refers to the K-nearest point of many other points in the high-latitude feature space and can damage detection performance.

In this paper, we explore two gaps in the few-shot setting for the support branch in meta-learning-based methods, which lead to instability and forgetting. To tackle these issues, we propose a simple but effective method named sample normalization to stabilize the meta-learning-based network and maintain consistent detection performance for both novel and base classes. We demonstrate the effectiveness of our method using Meta-DETR [18] as the backbone and apply sample normalization in the support branch. Additionally, we mitigate the negative impact of hubness by applying Z-score normalization [30] followed by a projection layer. Caused by the high-dimensional feature space, the hubness problem refers to the K-nearest point of many other points in the high-latitude feature space and can damage the performance. As shown in Figure 1c, with sample normalization, the network obtains more stable detection results for the novel class under different two-shot training samples. Experimental results show that our method presents competitive and stable performance and also reduces the problem of forgetting.

In summary, our main contributions are as follows:We reveal two gaps caused by few-shot settings in the meta-learning framework between different data, which hurt both performance stability for novel classes and the reduction of the results concerning the base classes.We propose a method named sample normalization under the meta-learning framework [18] to improve the ability of the network to migrate meta-knowledge to mitigate forgetting and enhance the stability of the network for different small samples.Experimental results demonstrate that our method achieves competitive performance for the detection of both novel classes and base classes over a single run and multiple experiments and can stabilize the network when facing various few-shot training objects.

## 2. Related Work

### 2.1. Object Detection

In recent years, as a fundamental computer vision task, plenty of work has been undertaken for object detection, which can be mainly divided into one-stage, two-stage, and transformer-based methods. Based on RCNN [32], two-stage algorithms first generate proposals and revise them to accomplish classification and bounding box regression. Then, the following detectors [7,8,33,34,35] gradually improve the detection speed and accuracy. In contrast, one-stage methods like YOLO [4] and SSD [36] directly encode the feature map to predict bounding boxes and categories, and they are applicable for real-time detection but obtain lower accuracy. The following algorithms [5,6,37,38,39] achieve a huge boost by adding different modules and training tricks. In addition, the transformer has aroused great attention in computer vision since ViT [40] first utilized a transformer in classification, and the following methods [41,42,43] demonstrated its superior performance in object detection. DETR [9] predicts the set containing all objectives at once and remains a simple architecture, achieving state-of-the-art performance while meeting convergence slowly. Deformable DETR [10] samples keys near the reference point to compute attention and solve the problem of DETR [9]. In contrast to one-stage and two-stage methods, transformer-based methods obtain a more concise structure and more representative features with better performance.

### 2.2. Few-Shot Object Detection

Few-shot object detection has many applications in practical scenarios. Existing work on few-shot object detection can be divided into four paradigms: data augmentation, distance metric learning, transfer learning, and meta-learning. Data augmentation [44] and distance metric learning [45] are less considered as solutions for FSOD. The former may not learn reliable hypotheses with a few data available. Distance metric learning is useful for classification but not for location. The mainstream methods are largely based on meta-learning and transfer learning. The transferability of features has been demonstrated in [26], which establishes the foundation for both approaches. A classical transfer-learning method, LSTD [11] is based on SSD [36] and Faster-RCNN [7] and trained under detection loss and regression loss to fine-tune the network. Furthermore, the following algorithms [15,16,17,44] further improve the accuracy with approaches like multiscale structure, contrastive learning, and so on.

Transfer-learning-based methods, by directly applying object detection knowledge from large data sets to a few-shot setting, can result in issues like class imbalance, overfitting, and complexity mismatch, thereby compromising generalization performance. Meta-learning approaches [19,20,21] have introduced the concept of ‘learning to learn’ into FSOD, which is achieved by extracting generalized knowledge across different meta-tasks. These approaches [24,25] utilize relation distillation and contrastive learning. By using prototypes [46] as the support branch, some meta-learning methods [47] thoroughly explore the relationship between the query branch and prototypes to address the FSOD problem. Meta-DETR [18] solves few-shot object detection at the image level, bypassing inaccurate region proposals and effectively handling multiple support categories simultaneously. Despite the FSOD tasks, some research [30,31] shows that high-dimensional features meet hubness, damaging detection performance. They propose to normalize the hyperspherical representations with zero mean, avoiding this problem.

The work published in recent years has primarily focused on improving detection accuracy through complex networks, with less emphasis on the stability of performance when dealing with various few-shot training samples. Our goal is to strike a balance between detection precision and stability within a meta-learning framework. We thoroughly examined the gaps that exist in the feature space for meta-learning-based methods under the few-shot setting, which have been overlooked in current research. Instead of adding complex modules, we propose a simple yet effective method. Regardless, we endeavor to alleviate the issue of forgetting while maintaining performance on novel classes, an aspect that has been neglected in most existing methods.

## 3. Materials and Methods

### 3.1. Overview and Motivation

In this section, we present the mathematical definition of few-shot object detection, the overview of the meta-learning framework, and our motivation.

**Problem definition:** Generally, there is a set of source domain $D_{s}$ containing base classes $C_{\mathrm{base}}$ with abundant labeled data for each class, while another set of target domain $D_{t}$ contains novel classes $C_{\mathrm{novel}}$, each of which has only a few annotated samples. We note that $C_{\mathrm{base}}\cap C_{\mathrm{novel}}=\emptyset$. If there are *C* classes in $C_{\mathrm{novel}}$ and each class only owns *K* training data, it is called a *C*-way *K*-shot object detection task.

**Overview:** The overall pipeline of the proposed framework is illustrated in Figure 2, where we use the well-known Meta-DETR [18] as the backbone, given a query image $I_{\mathrm{query}}=\left\{x_{\mathrm{query}}\right\}$ and a set of support images $I^{\mathrm{support}}=\left\{x_{k}^{s}\right\},k\in \left[1,N\right]$ with instance annotations, where N is the number of support samples once input to the net. $I^{\mathrm{support}}$ is from various randomly sampled classes, and only ground-truth objects belonging to the support classes are kept as input of the support branch. A weight-shared feature extractor first encodes them into the same feature space and obtains corresponding feature maps $F^{q}=\left\{f^{q}\right\}$ and $F^{s}=\left\{f_{k}^{s}\right\}$. A transformer encoder obtains the feature vector $S=\{s_{k}\}$ of each support sample, taking task encoding $T^{s}=\left\{t_{k}^{s}\right\}$ of each support object into account. Subsequently, we use another transformer with a correlation aggregation module to polymerize the support feature and the query feature, exploiting the interclass correlation and obtaining the enhanced query feature $f^{\mathrm{eq}}$. Finally, the proposed sample normalization is applied in the support branch to obtain a relatively consistent input and weaken the gaps, with aggregates Z-score normalization before final detection.

**Motivation:** The instability of the meta-learning-based network under different few-shot data limits its practical applicability. We deeply consider two gaps in selecting support images, which lead to instability and negative transfer of meta-knowledge, resulting in the forgetting problem.

**Gap between the sizes of training and test data:** FSOD faces a challenge due to the distribution gap between training and test data sets in the target domain. The training samples are mostly fixed without artificial choice, which are randomly oriented toward the test data. Meta-DETR randomly samples support images from the few-shot data set during meta-fine-tuning. As shown in Figure 3a, few-shot samples used for training must be randomly combined in the feature space. Sample A selects images close between classes but far from the center of the category within the class, located at the class edge in different directions. Sample B chooses objects far away between classes, with large intercategory distances and relatively close intraclass distances to the class center. These sampling combinations exhibit significant differences in the feature space. With limited combinations in the few-shot data sets, the weights of neural networks are biased towards labeled data, which may cause mismatching for test data. When facing various few-shot data sets, the network performance obtains poor stability.

**Gap between the sizes of the source and target domain:** In meta-learning frameworks, the randomness in selecting support data from the source domain, due to the difference in data volume between the source and target domains, hinders the migration of meta-knowledge and leads to forgetting. Figure 3a illustrates various randomly sampled support object combinations in the feature space, such as sampling A, sampling B, sampling C, etc. These combinations in the source domain often do not align with the fixed few combinations in the target domain’s feature space, causing the migration of meta-knowledge. This discrepancy leads to the target domain’s detection not fully utilizing the meta-knowledge learned in the source domain, which is detrimental to detection. Subsequently, after fine-tuning, the network tends to favor the few-shot combination in the support branch, leading to the forgetting of various combinations in the source domain.

These two types of gaps, arising from both large and small sample sizes, constitute the core challenge in few-shot learning. One randomness arises from the transition from small to large data, while the other results from the transition in the opposite direction. To address these dual randomness issues, we introduce sample normalization within the meta-learning framework for support objects. This normalization aims to minimize the disparities between the source and target domains, as well as to harmonize the disparities among training and testing samples within the target domain. As shown in Figure 3b, we standardize the support data onto a unit hypersphere, ensuring consistency in the combinations of input support samples. Through this normalization process, we aim to generalize support features during meta-training and meta-fine-tuning, thereby enhancing the transfer of meta-knowledge and mitigating the forgetting of base classes. Additionally, this approach helps narrow the divide between training and test data, further enhancing the model’s performance.

To address these gaps, we propose sample normalization (SN) for the meta-learning framework, specifically targeting support objects. This normalization aims to mitigate the challenges posed by the large and small sample sizes, thereby reducing the discrepancies between the source and target domains and the differences between training and testing samples in the target domain.

### 3.2. Sample Normalization

We implement the sample normalization (SN) on S to obtain $S^{\prime }=\left\{s_{k}^{\prime }\right\}$ as
(1)$$s_{k}^{\prime }=\frac{s_{k}-\mu }{\sigma },$$
where k∈[1,N], $\mu =\frac{1}{N}\sum_{k=1}^{N}s_{k}$, and $\sigma^{2}=\frac{\sum_{k=1}^{N}{\left(s_{k}-\mu \right)}^{2}}{N-1}$. This sample normalization is taken at each forward process, not at each episode, so this occurs for the distribution of the SN-passed support feature $S^{i}$ in the feature space in the *i*-th forward process. As shown in Figure 3b, we normalize the support data into a unit hypersphere and make combinations of input support samples standardized. The support features following SN $S^{\prime }$ reside on the identical unit hypersphere, ensuring both the relative positioning of support samples and the aggregation of samples of the same class in limited-data sets. This configuration offers dual benefits:(1)It mitigates the disparity in data volume between base training and fine-tuning stages. Specifically, meta-knowledge acquisition entails progressively learning to discern query images using randomly sampled support objects, as depicted in Figure 2. Due to data volume disparities between source and target domains, the support feature combinations encountered during base training often mismatch with the limited support features of few-shot samples. Regardless of the training stage, SN selects support samples that reside on the same hypersphere. This alignment reduces representation gaps across training stages, enhancing meta-knowledge transfer and maintaining base class detection performance.(2)SN mitigates inconsistencies within random few-shot training data within a fixed target domain. In practical applications, target domain data remain constant for a single experiment but may vary across multiple experiments. When training support samples for the same category are randomly selected from different images, network weights can diverge, affecting stability. By implementing SN during fine-tuning, the transformer’s input reduces intraclass disparities while maintaining relative sample characteristics beyond mere categorical centroids.

### 3.3. Z-Score Normalization

In FSOD, the similarity of input features in the feature space due to the small number of samples can lead to the hubness problem. To address it, we apply Z-Score normalization (ZN) followed by a linear projection before final detection. Concretely, let $f_{i}$, i∈[1,2,...,D] denote the i-th component of each enhanced query feature $f^{\mathrm{eq}}\in R^{D}$. Furthermore, we first apply Z-score normalization to obtain $f^{\mathrm{zn}}$
(2)$$f^{zn}=\frac{f^{eq}-\mu_{f}1}{\sigma_{f}}\in R^{D},$$
where $\mu_{f}=\frac{1}{D}\sum_{i=1}^{D}f_{i}$, $\sigma_{f}^{2}=\frac{1}{D}\sum_{i=1}^{D}{\left(f_{i}-\mu_{f}\right)}^{2}$, and $1={\left[1,1,1...,1\right]}^{T}$ is a D-dimensional vector with its components. Subsequently, we apply a linear layer to obtain the final feature f: (3)$$f=W\times f^{zn}+B$$
where W and B are parameters which we train in each episode.

ZN projects the original feature vectors along 1 to a hyperplane which contains the original and is perpendicular to 1. These vectors are then scaled to the same length of $\sqrt{D}$, which can be calculated by $∥f^{\mathrm{zn}}∥$.

### 3.4. Network Training

In this section, we provide the detailed loss functions utilized to optimize our network and training procedure for the meta-learning framework.

**Loss function:** Assume that our meta framework infers N predictions as the same as the fixed number of object queries within a single feed-forward process. Furthermore, we denote the query image by $x_{\mathrm{query}}$ and represent the ground-truth set of objects within $x_{\mathrm{query}}$ by $y={\left\{y_{i}\right\}}_{i=1}^{N}$. When $y_{i}$ indicates an object, $y_{i}=\left(c_{i},b_{i}\right)$, where $c_{i}$ denotes the class label and $b_{i}$ denotes the bounding box of the object. When $y_{i}$ indicates no object, $y_{i}=\left(\emptyset ,\emptyset \right)$. Our meta-framework makes N predictions for target class, which are $\hat{y}={\left\{\hat{y_{i}}\right\}}_{i=1}^{N}={\left\{\left(\hat{c_{i}},\hat{b_{i}}\right)\right\}}_{i=1}^{N}$. Our method follows deformable DETR [10], which adopts a set-based Hungarian loss that forces unique prediction for each object via bipartite matching. We adopt a pair-wise matching loss $L_{match}\left(y_{i}^{\prime },\hat{y_{\sigma (i)}}\right)$ to search for a bipartite matching between $\hat{y}$ and $y^{\prime }$ with the lowest cost:(4)$$\hat{\sigma }=\underset{\sigma }{argmin}\sum_{i=1}^{N}L_{match}\left(y_{i}^{\prime },\hat{y_{\sigma (i)}}\right),$$
where σ denotes a permutation of N elements and $\hat{\sigma }$ denotes the optimal assignment between predictions and targets. Since the matching should consider both classification and localization, the matching loss is defined as
(5)$$L_{match}\left(y_{i}^{\prime },\hat{y_{\sigma (i)}}\right)=\ell_{1_{\left\{c_{i}^{\prime }\neq \emptyset \right\}}}L_{cls}\left(c_{i}^{\prime },{\hat{c}}_{\sigma (i)}\right)+\ell_{1_{\left\{c_{i}^{\prime }\neq \emptyset \right\}}}L_{box}\left(b_{i}^{\prime },{\hat{b}}_{\sigma (i)}\right).$$

With the optimal assignment of $\hat{\sigma }$ obtained by Equations (Equation 4) and (Equation 5), we optimize the network using the following loss function: (6)$$L\left(y^{\prime },\hat{y}\right)=\sum_{i=1}^{N}\left[L_{cls}\left(c_{i}^{\prime },{\hat{c}}_{\sigma (i)}\right)+\ell_{1_{\left\{c_{i}^{\prime }\neq \emptyset \right\}}}L_{box}\left(b_{i}^{\prime },{\hat{b}}_{\sigma (i)}\right)\right],$$
where we use sigmoid focal loss [48] for $L_{cls}$ and apply a linear combination of $\ell_{1}$ loss and GIoU loss [49] for $L_{box}$. Furthermore, $L(y^{\prime },\hat{y})$ is employed to each layer of the transformer decoder. According to Meta-DETR [18], we utilize a cosine similarity cross-entropy loss [50] to classify the class prototypes of various classes to be distinguished from each other.

**Two-stage training strategy:** Our method follows the training strategy made by Meta-DETR [18], consisting of meta-training and meta-fine-tuning. In meta-training, the support image and the query image are selected from $D_{s}$ as the input of the two branches with abundant samples from each base class. When fine-tuning, only *K* objects (*K* shots) are available for every novel category. Several objects from base classes are selected with *K*-shots novel objects to form a data set as the input of the support and query branches.

## 4. Experiment

### 4.1. Experimental Settings

**Dataset:** Pascal VOC [2,3] is a widely used FSOD data set containing a large number of annotated images across 20 categories. The model is trained on the Pascal VOC 07+12 train-evaluation data and evaluated on the test data in VOC 07 [2]. We employ the settings of Meta-DETR [18] for both the source and target domain categories, a common configuration in FSOD. To evaluate our approach, we take three types of target and source domain categories into account. For instance, we use (‘bird’, ‘bus’, ‘cow’, ‘motorbike’, ‘sofa’/others), (‘aeroplane’, ‘bottle’, ‘cow’, ‘horse’, ‘sofa’/others), and (‘boat’, ‘cat’, ‘motorbike’, ‘sheep’, ‘sofa’/others) as our target domain/source domain categories. We varied the number of few-shot sample shots from 1 to 10. Mean average precision (mAP) and mean average recall (mAR) values at an IoU threshold of 0.5 are used to measure the performance. We use different samples for training, obtaining individual and average performance results across 10 groups of randomly sampled few-shot data sets.

**Implementation details:** We adopt Resnet-101 [51] as the feature extractor and maintain the same hyperparameters and network architecture as [10]. For fair comparisons, the model is implemented in a single-scale version and aggregated following [52]. Our model is trained on one Nvidia A6000 GPU, and the AdamW optimizer with initial parameters is adopted. The batch size and the number of support objects are set to 16 and 5, respectively. For Pascal VOC, the initial learning rate of $2\times 10^{-4}$ and a weight decay of $1\times 10^{-4}$ are performed for 50 epochs of meta-training. Subsequently, we use the same settings as [18] to fine-tune the model on the few-shot data set until convergence and retain the best results.

### 4.2. Results and Discussion

**Detection performance for novel classes:** Table 1 presents the FSOD performance (mAP@0.5) for novel classes of Pascal VOC *test07*, which includes results from both single-run and multiple-run experiments. We conducted 10 runs with different random seeds, each utilizing different few-shot samples from the three class splits. For fair comparisons, we retrain Meta-DETR with the original settings denoted by *. This adjustment achieved better performance than results in [18]. As shown in Table 1, our method outperforms others in most *K*-shot settings across class splits in a single experiment. With sample normalization and Z-score normalization, we observe noticeable performance improvements with an increase of up to +0.3 mAP@0.5 in 1 shot, +4.4 mAP@0.5 in 2 shots, +3.8 mAP@0.5 in 3 shots, +1.7 mAP@0.5 in 5 shots, and +3.0 mAP@0.5 in 10 shots. It proves our method can effectively enhance the detection capability of the meta-learning framework, particularly for larger *K*.

Across multiple runs with randomly sampled support objects, our method outperforms the original framework in average mAP@0.5, particularly in class splits 1 and 3, with a notable margin of +3.6 mAP@0.5 in 2 shots, +4.8 mAP@0.5 in 3 shots, +1.5 mAP@0.5 in 5 shots, and +2.8 mAP@0.5 in 10 shots. Although our method may show slightly lower performance in split 2 and for lower shots in split 1 and 3, it still maintains strong detection ability compared to Meta-DETR. Table 1 demonstrates the superiority of our method in FSOD tasks, even with various few-shot training samples. For mAR, our method consistently outperforms the baseline, with improvements of up to +5.1 mAR in 1 shot, +3.9 mAR in 2 shots, +4.7 mAR in 3 shots, +3.2 mAR in 5 shots, and +3.7 mAR in 10 shots in a single run. Moreover, over multiple runs, our method achieves the following improvements: up to +4.8 mAR, +4.4 mAR, +5.1 mAR, +1.8 mAR, and +3.2 mAR, respectively, in 1, 2, 3, 5, and 10 shots, as shown in Table 2.

**Detection stability:** We analyze the stability of our method via variance of detection performance, including mAP@0.5 and mAR, as shown in Table 3. Our method exhibits smaller variance in most settings, indicating improved stability. We can see that our method is particularly effective in stabilizing mAR.

We plot the scatter of results (mAP@0.5 and mAR) in Figure 4 for 3 class splits. We can conclude that our method makes the performance more clustered facing various few-shot objects from (a)–(j), although there exist several outliers causing the increase of variance, specifically mAP@0.5 in one shot and two shots of the first class split and in five shots of the second class split and so on.

In Figure 5 and Figure 6, we show our proposed sample normalization demonstrates superior detection performance across different two-shot training samples in class split 3. Additionally, even when faced with various changes in few-shot training samples, our method exhibits greater stability compared to the baseline [18], fully showcasing the enhancement of sample normalization on the stability of the meta-learning framework.

**Detection performance for base classes:** Our method demonstrates superior performance in both base and novel classes compared to other methods, as indicated in Table 4. Additionally, we show significant progress in reducing forgetting compared to the baseline, Meta-DETR. Specifically, we obtain +11.2 mAP@0.5 in 1 shot, +13.7 mAP@0.5 in 3 shots, +13.8 mAP@0.5 in 5 shots, and +15.0 mAP@0.5 in 10 shots, while we remain mostly the best performance in novel classes. Although there is a limited decrease in mAP@0.5 after fine-tuning, our method retains effective base class distinction. Our proposed sample normalization with Z-score normalization consistently improves FSOD performance both in single runs and across multiple experiments. It enhances the stability of the meta-learning network when trained on diverse few-shot samples while maintaining superior performance for both base and novel classes.

**Dense object detection:** We were pleasantly surprised to find that our method performs exceptionally well in detecting dense targets, as demonstrated in Figure 7. Specifically, in challenging scenarios involving dense clustering, variable target sizes, or occlusions, our approach consistently outperforms the baseline, accurately identifying individual targets without missing any or falsely detecting entire clusters.

### 4.3. Ablation Study

In Table 5, we present the results (mAP@0.5 and mAR) of the network with and without sample normalization and Z-score normalization for novel classes in the third class split of Pascal VOC. Sample normalization maintains mAP@0.5 with only a slight decrease in most settings and demonstrates improved performance in three shots over multiple runs. It effectively enhances mAR for novel classes in both single and multiple experiments. Additionally, Z-score normalization improves mAP@0.5 in lower shots and shows a slight enhancement in mAR performance for novel classes. When both SN and ZN are used, our method often achieves a balanced performance across all aspects.

Furthermore, as shown in Table 6, we present the detection results for base classes in a class split 3 of Pascal VOC. We can draw the conclusion that SN can effectively solve the problem of forgetting for Meta-DETR both in mAP and in mAR and can nearly retain the ability to detect base classes when the number of shots is greater than one. For instance, before meta-fine-tuning, the mAP@0.5 for base classes is 92.6 and the mAR is 77.1.

We plot the results of mAP@0.5 and mAR for the third class split in Pascal VOC [2,3], respectively, in Figure 4k,l. As illustrated in Figure 4k,l, we can draw the conclusion that our proposed sample normalization does affect the stability of the network and cluster the results in each shot.

In Figure 4k, there exist two outliers in one shot, representing the best and the worst results. Furthermore, we show the on3-shot training samples corresponding to these results in Figure 8. These two randomly selected targets exhibit extreme differences, with the first row achieving the best mAP@0.5 of 43.5, while the second row only achieves 20.6. We analyze that the reason for this, which is that the sample normalization method causes the network to normalize relatively uniform samples in the feature space, without disrupting the relative positions.

## 5. Conclusions

Addressing the issue of instability and forgetting in the meta-learning framework, we deeply analyze two gaps: the first one is the gap between the sizes of training data and test data, which leads to instability, and the second is the gap between the sizes of the source domain and target domain, causing forgetting. We propose sample normalization applied in the support branch, fusing Z-score normalization to solve the problem of hubness, which obtains better performance especially compared with the baseline, i.e., Meta-DETR. Using two universal metrics, mAP and mAR, we prove that our method can not only stabilize Meta-DETR when facing various few-shot samples but also partly maintain the detection ability for base classes.

**Limitations:** Although our method has strong capabilities in stabilizing meta-learning frameworks and reducing source domain forgetting, in some extreme cases, such as the significant differences in samples under different random numbers mentioned above, the network always tends to favor one extreme, unable to balance the detection performance in both situations. Additionally, compared to some algorithms [47] that do not require fine-tuning and can be directly used for detecting new categories, the two-stage training strategy is quite complex and time-consuming. In the future, we will continue to explore more effective training methods and research on sample data distribution in order to eliminate the gap from few-shot to large-sample data.

## Figures and Tables

**Figure 1 sensors-24-03456-f001:**
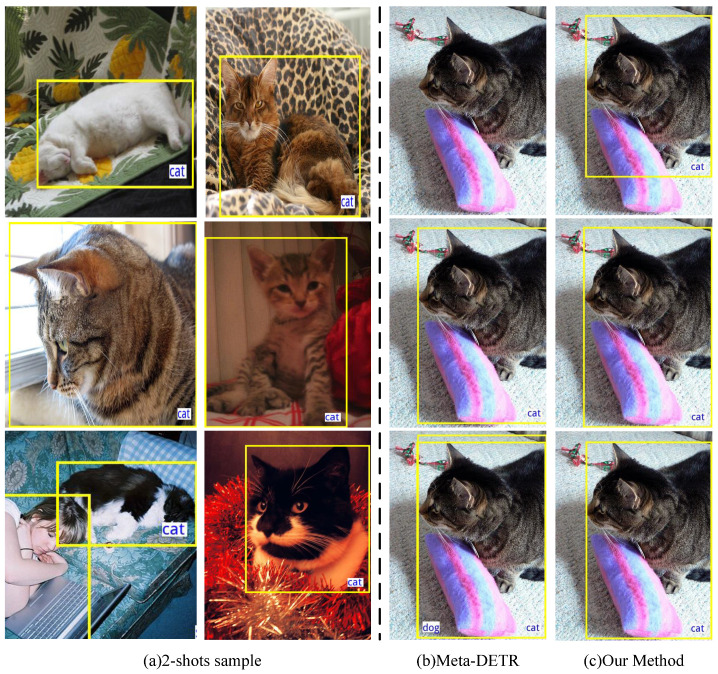
Few-shot training samples and corresponding object detection results. Each row is under a seed to select two-shot training samples: (**a**) presents two-shot training samples of a novel class ‘cat’; (**b**,**c**) reveal the results of detecting ‘cat’ using Meta-DETR [18] and our method, respectively. In (**b**), under various few-shot samples, Meta-DETR recognizes nothing in row 1 and misidentifies the ‘cat’ as ‘dog’ in row 3, while our method can achieve stable performance, as shown in (**c**).

**Figure 2 sensors-24-03456-f002:**
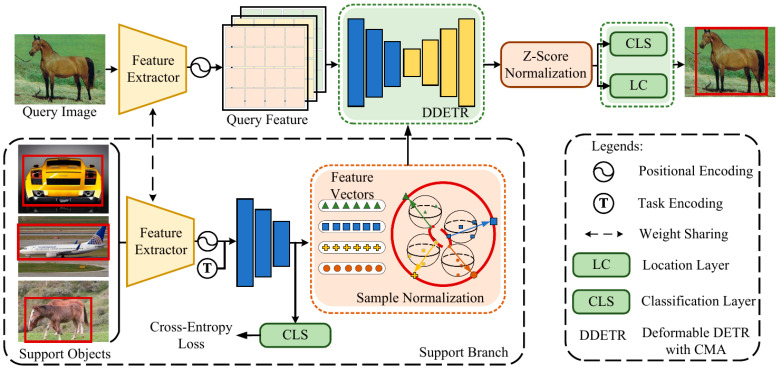
Overall pipeline of our framework, where we use Meta-DETR [18] as the backbone. Sample normalization is applied after translating each support image into an embedding vector, and Z-score normalization is used before the final detection.

**Figure 3 sensors-24-03456-f003:**
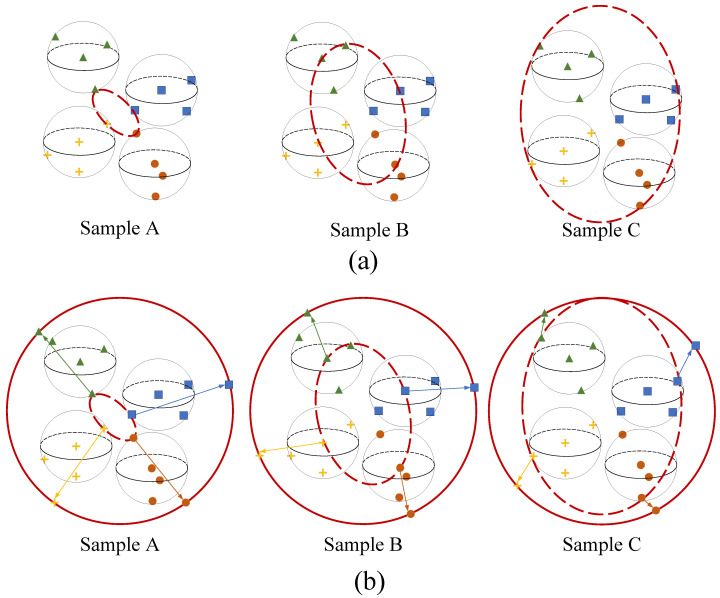
Multiple sampling strategies in feature space for support branch in the meta-learning framework. A hypersphere in the feature space represents a category, and different points in one red circle represent support images sampled in each episode for each category. We only display three sample combinations of A, B, and C. As shown in (**a**), without sample normalization, each support object is randomly sampled for its category and combined into random shapes in the feature space like sample A, sample B, and sample C. The images in (**b**) represent combinations of support objects after sample normalization, which are distributed on a standard hypersphere.

**Figure 4 sensors-24-03456-f004:**
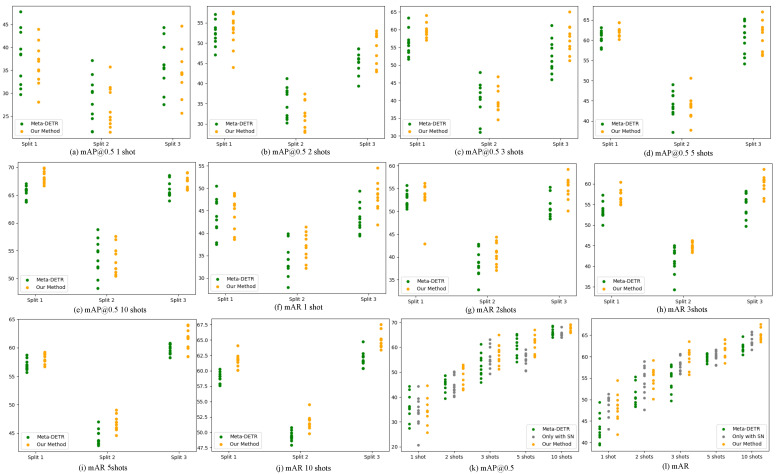
Results dispersion for different shots under 10 seeds in Pascal VOC [2]. (**a**–**e**) plot the mAP@0.5 in 1, 2, 3, 5, and 10 shots, respectively. (**f**–**j**) present scatter for mAR in 1, 2, 3, 5, and 10 shots, respectively, which are both for 3 class splits. In (**k**,**l**), we show the dispersion contrast between the Meta-DETR [18] with/without our proposed sample normalization.

**Figure 5 sensors-24-03456-f005:**
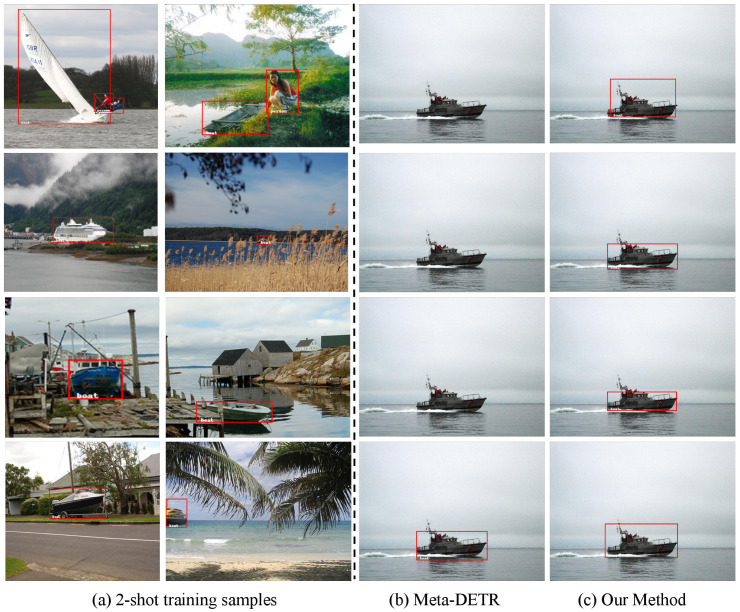
Few-shot training samples in class split 3 and corresponding object detection results for the novel class ‘boat’. Each row is under a seed to select 2-shot training samples. (**a**) presents 2-shot training samples of a novel class ‘boat’. (**b**,**c**) reveal the results of detecting ‘boat’ using Meta-DETR [18] and our method, respectively. In (**b**), under various few-shot samples, Meta-DETR recognizes nothing in rows 1–3 and provides a somewhat inaccurate bounding box in row 4, while our method can obtain stable performance as shown in (**c**).

**Figure 6 sensors-24-03456-f006:**
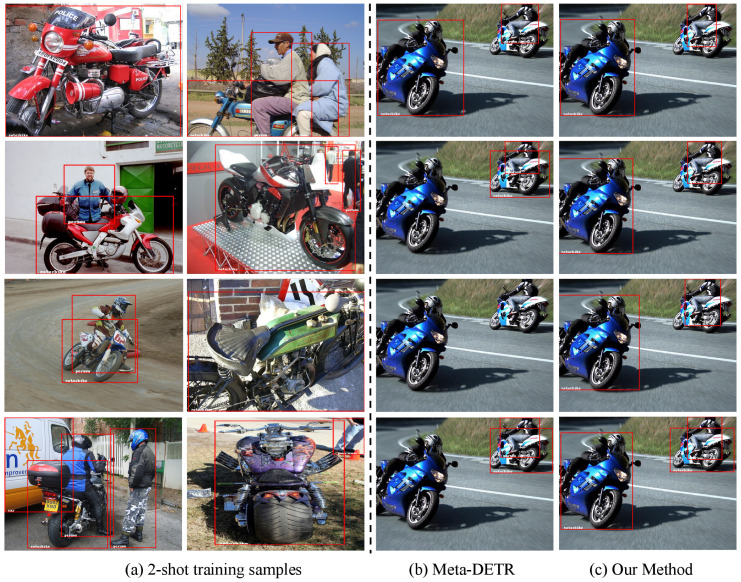
Few-shot training samples in class split 3 and corresponding object detection results for the novel class ‘motorbike’. Each row is under a seed to select 2-shot training samples. (**a**) presents 2-shot training samples of a novel class ‘motorbike’. (**b**,**c**) reveal the results of detecting ‘motorbike’ using Meta-DETR [18] and our method, respectively. In (**b**), under various few-shot samples, Meta-DETR shows significant variance, while our algorithm demonstrates excellent stability and improved detection performance across different training samples as shown in (**c**).

**Figure 7 sensors-24-03456-f007:**
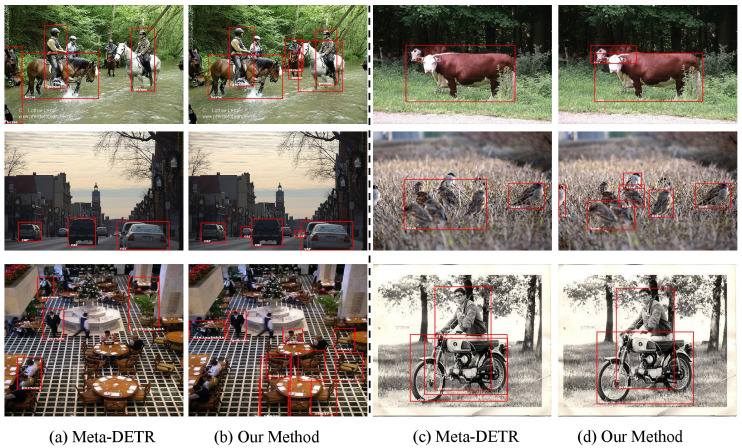
Comparison of detection results for occluded or densely packed objects. (**a**,**c**) show the detection results of Meta-DETR, while (**b**,**d**) correspond to the detection results of our method for the left images.

**Figure 8 sensors-24-03456-f008:**
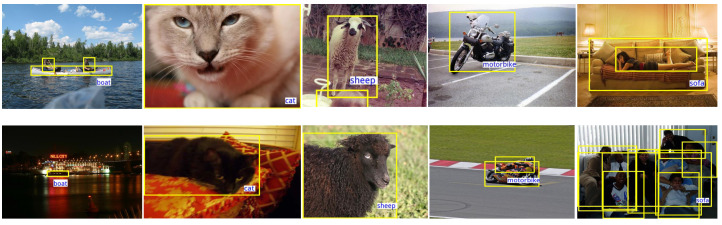
One-shot training objects for class split 3 in Pascal VOC [2,3]. The images in row 1 and row 2 are randomly sampled from the same new classes using different random seeds. With vastly different training samples, Meta-DETR with the proposed sample normalization yields two extreme performances, achieving the best results (mAP@0.5 43.5) in row 1 and the inferior results (mAP@0.5 20.6) in row 2. We mark the bounding boxes and categories of novel classes, while only the bounding boxes of the base classes are marked. Some images are clipped for visualization.

**Table 1 sensors-24-03456-t001:** Quantitative comparisons (mAP@0.5) on Pascal VOC *test07* [2] for novel classes. We tested the results of a single experiment of 1, 2, 3, 5, and 10 shots in 3 class splits. ‘Multiple’ represents that results are averaged over 10 random runs, while ‘Single’ is over a single run. The samples for these 10 experiments were sampled based on 10 random numbers. The best results are highlighted in bold, and the second-best results are underlined. * denotes our reimplemented results.

	Method/Shots	Class Split 1					Class Split 2					Class Split 3				
		1	2	3	5	10	1	2	3	5	10	1	2	3	5	10
Single	LSTD [11]	8.2	1.0	12.4	29.1	38.5	11.4	3.8	5.0	15.7	31.0	12.6	8.5	15.0	27.3	36.3
	TFA w/cos [12]	39.8	36.1	44.7	55.7	56.0	23.5	26.9	34.1	35.1	39.1	30.8	34.8	42.8	49.5	49.8
	MPSR [44]	41.7	43.1	51.4	55.2	61.8	24.4	29.5	39.2	39.3	47.8	35.6	40.6	42.3	48.0	49.7
	Retentive R-CNN [29]	42.4	45.8	45.9	53.7	56.1	21.7	27.8	35.2	37.0	40.3	30.2	37.6	43.0	49.7	50.1
	TFA w/cos+Halluc [53]	45.1	44.0	44.7	55.0	55.9	23.2	27.5	35.1	34.9	39.0	30.5	35.1	41.4	49.0	49.3
	CME [23]	41.5	47.5	50.4	58.2	60.9	27.2	30.2	41.4	42.5	46.8	34.4	39.6	45.1	48.3	51.5
	SRR-FSD [13]	**47.8**	50.5	51.3	55.2	56.8	32.5	35.3	39.1	40.8	43.8	40.1	41.5	44.3	46.9	46.4
	FSCE [15]	44.2	43.8	51.4	61.9	63.4	27.3	29.5	43.5	44.2	50.2	37.2	41.9	47.5	54.6	58.5
	Meta-DETR [18] *	47.7	57.1	63.3	63.1	67.1	**37.1**	**41.2**	43.6	**49.1**	54.6	44.3	48.6	61.2	65.3	68.6
	Ours	43.9	**57.3**	**64.0**	**64.6**	**69.8**	35.7	37.4	**46.7**	49.0	**57.6**	**44.6**	**53.0**	**65.0**	**67.0**	**69.1**
Multiple	Deformable-DETR-ft-full [10]	5.6	13.3	21.7	34.2	45.0	10.9	13.0	18.4	27.3	39.4	7.3	16.6	20.8	32.2	41.8
	TFA w/cos [12]	25.3	36.4	42.1	47.9	52.8	18.3	27.5	30.9	34.1	39.5	17.9	27.2	34.3	40.8	45.6
	FsDetView [52]	24.2	35.3	42.2	49.1	57.4	21.6	24.6	31.9	37.0	45.7	21.2	30.0	37.2	43.8	49.6
	MPSR [44]	34.7	42.6	46.1	49.4	56.7	22.6	30.5	31.0	36.7	43.3	27.5	32.5	38.2	44.6	50.5
	DCNet [24]	33.9	37.4	43.7	51.1	59.6	23.2	24.8	30.6	36.7	46.6	32.3	34.9	39.7	42.6	50.7
	FSCE [15]	32.9	44.0	46.8	52.9	59.7	23.7	30.6	38.4	43.0	48.5	22.6	33.4	39.5	47.3	54.0
	Meta-DETR [18] *	**37.8**	52.3	56.1	60.7	65.5	**28.5**	**35.3**	**40.2**	**44.1**	**53.7**	**36.1**	45.2	52.6	60.4	66.0
	Ours	36.3	**52.8**	**59.7**	**62.2**	**68.3**	27.1	32.2	39.7	43.3	53.3	34.5	**48.8**	**57.4**	**61.2**	**67.4**

**Table 2 sensors-24-03456-t002:** Quantitative comparisons (mAR) for the Meta-DETR [18] and our method on Pascal VOC [2]. The best results are highlighted in bold. * denotes our reimplemented results.

	Method/Shots	Class Split 1					Class Split 2					Class Split 3				
		1	2	3	5	10	1	2	3	5	10	1	2	3	5	10
Single	Meta-DETR *	47.6	55.7	57.3	58.7	60.3	39.9	42.8	45.0	47.0	50.8	49.4	55.3	58.2	60.8	64.7
	Ours	**48.9**	**56.2**	**58.5**	**59.2**	**62.4**	**41.4**	**44.4**	**46.1**	**49.1**	**54.5**	**54.5**	**59.2**	**63.5**	**64.0**	**67.5**
Multiple	Meta-DETR *	43.7	52.7	53.4	57.0	58.9	33.9	38.5	41.4	44.3	49.4	43.4	50.8	54.8	59.9	61.9
	Ours	**44.6**	**52.9**	**56.9**	**58.2**	**61.6**	**36.9**	**40.7**	**44.9**	**46.7**	**51.7**	**48.2**	**55.2**	**59.9**	**61.7**	**65.1**

**Table 3 sensors-24-03456-t003:** The variance of the mean mAP@0.5 of 10 seeds on Pascal *test07* [2] for Meta-DETR [18] and our method. Lower is better, and the best results are highlighted in bold.

Method/Shots	Class Split 1					Class Split 2					Class Split 3				
	1	2	3	5	10	1	2	3	5	10	1	2	3	5	10
Meta-DETR	33.7	8.2	11.9	2.7	1.3	24.5	13.7	24.8	10.3	10.0	26.0	7.5	20.8	13.3	2.1
Ours	**19.4**	16.1	**3.8**	**1.8**	**1.1**	**19.6**	**10.1**	**12.2**	**9.9**	**6.1**	**25.1**	13.9	**15.7**	**12.1**	**1.3**

**Table 4 sensors-24-03456-t004:** Quantitative comparisons (mAP@0.5) for both base and novel classes on the first split of Pascal VOC [2]. The best results are highlighted in bold, and the second-best results are underlined. ^†^ indicates results averaged on multiple runs.

Method/Shots	Base Classes				Novel Classes			
	01	03	05	10	01	03	05	10
Meta-YOLO [20]	66.4	64.8	63.4	63.6	14.8	26.7	33.9	47.2
FsDetView [52] ^†^	64.2	69.2	69.8	71.1	24.2	42.2	49.1	57.4
TFA w/cos [12] ^†^	77.6	77.3	77.4	77.5	25.3	42.1	47.9	52.9
MPSR [44] ^†^	60.6	65.9	68.2	69.8	34.7	46.1	49.4	56.7
FSCE [15] ^†^	75.5	73.7	75.0	75.2	32.9	46.8	52.9	59.7
Meta-DETR ^†^	67.2	70.0	73.0	73.5	**37.8**	56.1	60.7	65.5
Ours ^†^	**78.4**	**83.7**	**86.8**	**88.5**	36.3	**59.7**	**62.2**	**68.3**

**Table 5 sensors-24-03456-t005:** Ablation results (mAP@0.5) for novel classes in class split 3 on Pascal VOC 07 [2]. ‘Multiple’ represents that results are averaged over multiple random runs, while ‘Single’ is over a single run.

	SN	ZN	mAP@0.5					mAR				
			01	02	03	05	10	01	02	03	05	10
Single			44.3	48.6	61.2	65.3	68.6	49.4	55.3	58.2	60.8	64.7
		✓	53.6	49.8	56.5	64.3	67.0	55.1	59.0	58.0	61.0	64.7
	✓		43.5	50.2	61.5	59.1	68.2	51.3	58.9	60.6	61.6	65.2
	✓	✓	44.6	53.0	65.0	67.0	69.1	54.5	59.2	63.5	64.0	67.5
Multiple			36.1	45.2	52.6	60.4	66.0	43.4	50.8	54.8	59.9	61.9
		✓	41.8	45.7	51.1	58.7	65.8	48.8	53.1	55.1	59.5	62.8
	✓		34.0	44.7	55.9	55.0	65.2	48.8	54.2	58.0	60.1	63.7
	✓	✓	34.5	48.8	57.4	61.2	67.4	48.2	55.2	59.9	61.7	65.1

**Table 6 sensors-24-03456-t006:** Ablation results (mAP@0.5 and mAR) in a single run for base classes in class split 3 of Pascal VOC 07 [2].

SN	ZN	mAP@0.5					mAR				
		01	02	03	05	10	01	02	03	05	10
		63.1	75.9	77.1	85.9	85.4	55.2	63.3	62.8	70.4	70.6
✓		82.3	88.3	87.0	90.6	90.9	67.7	73.3	72.4	75.6	75.1
✓	✓	78.8	87.9	88.0	89.4	90.4	67.9	73.0	72.4	74.4	74.5

## Data Availability

The original contributions presented in the study are included in the article, further inquiries can be directed to the corresponding author.

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
