# Peer review of "Towards Stabilized Few-Shot Object Detection with Less Forgetting via Sample Normalization"

_sensors, 2024, doi:10.3390/s24113456_

Round 1

Reviewer 1 Report

Comments and Suggestions for Authors

The authors proposed a method for enhancing the performance stability of Meta-DETR in object recognition. The paper is well written, well organized and the reported results are very encouraging. However, I would like to make the following comments:

1. All acronyms used must have their meanings described when it is appearing the first time in the text. Meta-DETR, for example, is not explained.

2. Given the explosion of new works on the topic investigated in the last three years, the authors need to explain in more detail the novelties of their proposal in relation to what already exists in the literature.

3. I did not find the work of “Jason et al.”, mentioned in subsection 2.2, in the references.

4. Subsections 3.1 to 3.3 could be merged to form a single subsection.

5. Regarding the results, I missed a discussion establishing the relationships between the strengths/weaknesses of the proposed method and the characteristics of the objects/scenes contained in the images. This would help to better evaluate the suitability of the proposed method for different contexts and practical applications.

Reviewer 2 Report

Comments and Suggestions for Authors

The authors proposed a technique called sample normalization within a meta-learning framework to improve the network's ability to transfer meta-knowledge for mitigation.

All the analytical calculations set out in formulas (1-6) seem correct to me, although they do not explain the main advantages that the authors managed to achieve, namely why the program of the authors of the article has become simpler and more accurate. However, in all the tables given by the authors it is clear that the method proposed by the authors gives the best results.

Interesting results are shown in 6 tables and 8 figures. Everywhere the authors' results give better results than the closest Meta-DETR technique. Why this happens is hidden in artificial intelligence technology.

In tasks using self-learning and artificial intelligence, repeatability of results is important. The authors again show in Table 3 that their method exhibits less variance in most parameters, indicating improved stability.

The original results, presented in a sufficient number of tables, inspire confidence in their diversity in approaches to assessing the effectiveness of the proposed method, although, I repeat, neither I nor anyone else, sometimes except the authors, have the opportunity to directly verify them. This is an inherent property of problems solved using artificial intelligence! No one can reliably know “what this artificial intelligence “thought” at one moment or another, what it learned during the training stage”? That is why I so briefly agreed with the results presented in 6 tables and 8 figures explaining them. True, I had doubts about the correctness of using the Meta-DETR algorithm in Figure 5b, which is what I wrote about.

Only the result shown in Figure 5(b) seems strange. Indeed, it is very easy to identify a figure on a flat field, but Meta-DETR cannot cope. But the method proposed by the authors works well (Figure 5(c)).

I consider all references to previous works to be correct and I do not see any abuse, the introduction provides sufficient background information and includes all relevant references. Also, all cited references are directly related to the research of the authors of the presented article. I cannot give recommendations for improving this article, since I fundamentally do not deal with the problems of artificial intelligence and learning programs. For a physicist, and I am a physicist, it is much more important to find the physical nature of a phenomenon. This fundamentally cannot be done using artificial intelligence methods. Although some of the results obtained with its help can be very useful, as I wrote about in my review of the presented article.

I wrote that although, at first glance, the sample normalization method proposed by the authors seems simple and obvious, it gives good results. Therefore, the article by the authors Yang Ren & Co deserves attention, meets the requirements of Sensors and can be published without criticism.

I confirm everything that was written earlier and believe that there cannot be a more detailed analysis without a complete repetition of the work of the artificial intelligence program with sample normalization.
